biochemistry/materials science

microbiologically influenced corrosion, sulfate-reducing bacteria, stainless steel, hydrophobicity

**Author for correspondence:**
K. Kannoorpatti
e-mail: krishnan.kannoorpatti@cdu.edu.au

This article has been edited by the Royal Society of Chemistry, including the commissioning, peer review process and editorial aspects up to the point of acceptance.

# A study of bacteria adhesion and microbial corrosion on different stainless steels in environment containing *Desulfovibrio vulgaris*

T. T. T. Tran[1], K. Kannoorpatti[1], A. Padovan[2] and S. Thennadil[1]

[1]Energy and Resources Institute, College of Engineering, Information Technology and Environment, and [2]Research Institute for the Environment and Livelihoods, College of Engineering, Information Technology and Environment, Charles Darwin University, Darwin, Northern Territory 0909, Australia

TTTT, 0000-0003-4656-5018

Stainless steel is an important material used in many applications due to its mechanical strength and corrosion-resistant properties. The high corrosion resistance of stainless steel is provided by the passive film. Different stainless steels have different alloy elements and surface properties which could have a significant influence on bacterial attachment to the surface and thus might result in different microbial corrosion behaviours. In this study, the effect of adhesion of sulfate-reducing bacteria (SRB) on corrosion behaviour in artificial seawater on different stainless steels was investigated. Stainless steel materials used were SS 410, SS 420, SS 316 and DSS 2205 and pure chromium. The contact angle was measured to study the effect of surface properties of materials. Adhesion was measured by counting cells attached to the surface of materials. The corrosion behaviour of the materials was measured by electrochemical testing including measuring open circuit potential, electrochemical impedance spectroscopy and potentiodynamic behaviour. The long-term corrosion behaviour of each material was studied after six months of exposure by measuring weight loss and surface analysis with scanning electron microscope with energy-dispersive X-ray analysis. Hydrophobicity had a strong effect on bacterial attachment. Alloying elements e.g. nickel also had shown its ability to attract bacteria to adhere on the surface. However, the corrosion rate of different materials is determined not only by bacterial attachment but also by the stability of the passive film which is determined

by the alloying elements, such as Mo and Cr. Chromium showed high resistance to corrosion, possibly due to toxicity on bacterial attachment. The nature of bacterial attachment and corrosion behaviour of the materials are discussed.

## 1. Introduction

Microbiologically influenced corrosion (MIC), which results in deteriorating materials, is a serious problem in various industries [1–3] and can occur in oxic as well as in anoxic environments (e.g. marine sediments, deep seawater, water-logged soil). Sulfate-reducing bacteria (SRB) are one of the predominant types of bacteria associated with MIC [4,5]. In the absence of oxygen or low oxygen conditions, SRB converts sulfate ions to sulfides and can corrode metals through a series of oxidation and reduction reactions [1,6]. The corrosion deterioration of metals is primarily due to the sulfide ions [7,8] which can form metal sulfides and deteriorate the passive film, thus resulting in corrosion [6,9].

Generally, the adhesion process is considered as the precursor step to MIC of stainless steel. In this step, planktonic cells are attracted to the material surface by van der Waals and Coulomb interactions [10,11]. Then bacteria produce exopolymeric substances (EPS) to strengthen their attachment to the metal surface. In the subsequent stage, bacteria multiply to form a biofilm [12]. High bacterial adhesion may make the materials susceptible to corrosion. Thus, factors that impact on the attachment of bacteria to the material surface and the corrosion process should be taken into account to study MIC. There are many factors that affect the adhesion process, including bacteria characteristics, material surface and environmental factors [13]. It has been reported that both physical characteristics including surface roughness, surface tension, hydrophobicity and chemical composition of the material can influence bacterial adhesion [14].

Stainless steel including duplex stainless steel type is reported to be susceptible to MIC and pitting [15–21]. Due to the difference in surface properties and alloying elements, the adhesion of bacteria to the surfaces might vary, thus might result in differences in corrosion behaviour of the materials. Alloying elements such as nickel, chromium and nitrogen have been added to stainless steel in order to achieve specific properties. These alloying elements could significantly affect the attachment of bacteria and biofilm development and play an important role in the adhesion and corrosion process of materials in MIC environments [22]. Recent studies have shown the effect of several alloys on the adhesion of bacteria [14,23–33]. Nickel has been reported to enhance bacterial attachment on stainless steel surfaces as it increases the number of bacteria and colonies on the surfaces of materials in both aerobic and anaerobic environments [14,26,27]. Nitrogen was also reported to increase bacteria deposits on steel [29]. The authors claimed that the adhesion area on high nitrogen steel was higher than on stainless steel 304. On the contrary, molybdenum and ruthenium were shown to reduce bacterial attachment [26] even at low concentrations in the stainless steel alloys. Other antibacterial alloys such as copper and silver have also been investigated [30–33].

The addition of chromium to steel increases resistance to corrosion yet its effect on bacterial adhesion has received less attention. Previous studies have shown that chromium can be released into the environment as Cr (III) or Cr (VI) in both *in vivo* and *in vitro* environments [34,35]. Cr (VI) is known to be very toxic to microorganisms and can cause serious damage to the structure and diversity of microbial communities [36,37], such as decreased growth rate of bacteria and extended lag phase [38,39]. Cr (III), which is less toxic than Cr (VI), has been reported to decrease sulfate-reduction activity of SRB [38]. Thus, higher concentrations of chromium ions released into the environment might have a negative effect on microbial activities including reducing microbial adhesion on the surface of materials. Hydrophobicity is another factor that also has a significant impact on bacterial attachment and the hydrophobicity varies among materials. There is a lack of literature that shows the combined effect of hydrophobicity and alloying elements on bacterial adhesion to the surface of materials.

*Desulfovibrio vulgaris* is a species of Gram-negative sulfate-reducing bacteria in the Desulfovibrionaceae family [40] capable of corroding materials by hindering the passivation of the passive oxide layer of stainless steel. The existence of biofilms on metal surfaces frequently creates new electrochemical reaction pathways or allows reactions that are normally not favoured in the absence of microorganisms, which leads to increased corrosion. The metabolic products of bacteria can also significantly modify the interfacial processes between the biofilm and a metallic sublayer.

This present work is a study of microbial adhesion and corrosion behaviour in an artificial seawater environment containing *D. vulgaris*. This study aims to provide the factors that affect the adhesion

process of bacterial to the stainless steel surface and the corrosion mechanism of stainless steels with different added alloying elements, especially chromium in a corrosive microbial environment. Different materials were used in the study including a pure chromium plate in order to compare the influence of chromium in bacterial attachment and corrosion processes in stainless steels.

# 2. Material and methods

## 2.1. Materials

Stainless steel SS 410, SS 420, SS 316, DSS 2206 and high-purity chromium (99.99%) were used in this study. These are popular stainless steels with varying chromium contents and different crystal structures. The chemical composition of each type of stainless steel was determined by energy-dispersive X-ray fluorescence spectrometry (EDX-8100), and the composition is given in table 1. Chromium was the major alloying element providing corrosion resistance along with molybdenum that provides enhanced pitting resistance while nickel increases the hardenability of the steels.

To assess contact angle, adhesion and corrosion rates, coupons from each type of stainless steel ($10 \times 10 \times 2$ mm) were polished to 1 µm finish. After polishing, the coupons were rinsed with water, degreased with acetone, rinsed with distilled water, immersed in 80% ethanol for 2 h and finally dried in a biohazard cabinet to prevent any bacterial contamination before the experiments.

For electrochemical testing, each stainless steel coupon was mounted in a mould of non-conducting epoxy resin connected to an insulated copper wire to act as a working electrode. For long-term corrosion studies, coupons were immersed for six months. A same set of specimens was prepared for control experiments for electrochemical tests.

## 2.2. Medium and test conditions

Coupons were immersed in nutrient-rich artificial seawater consisting of modified Baar's medium (g l$^{-1}$): $MgSO_4$ 0.2; sodium citrate 0.5; $CaSO_4$ 0.1; $NH_4Cl$ 0.1; $K_2HPO_4$ 0.05; sodium lactate 3.5; yeast extract 1, added to 1 l of artificial seawater prepared according to ASTM 114-98 (g l$^{-1}$) [41]: NaCl 24.53; $MgCl_2$ 5.2; $Na_2SO_4$ 4.09; $CaCl_2$ 1.16; KCl 0.695; $NaHCO_3$ 0.201; KBr 0.101; $H_3BO_3$ 0.027; $SrCl_2$ 0.0025, NaF 0.003, in high pure water.

The pH of the test medium was adjusted to 7.4 using 1 M hydrochloric acid or 1 M sodium hydroxide then stirred for 30 min. The test medium was purged with nitrogen gas for 1 h and sterilized by autoclaving for 15 min at 121°C. Control condition was prepared the same as a biotic condition but without the presence of bacteria.

*Desulfovibrio vulgaris* (ATCC® 7757) (In Vitro Technologies, VIC, Australia), a species of SRB, was used in this study. The strain was stored at −80°C in 15% glycerol. Bacteria were retrieved from storage and cultured in 500 ml modified Baar's medium for 48 h at 37°C under anaerobic conditions. After approximately 48 h, 10 ml of bacteria culture medium was removed for determining bacterial concentration. The bacterial cells were harvested by centrifugation for 10 min, resuspended in 10 ml high pure water, stained with 0.4% trypan blue and counted using a haemocytometer. Five millilitres of culture medium was added to each 500 ml glass bottle containing nutrient-rich artificial seawater to give a final bacterial concentration of approximately $3.19 \times 10^4 \pm 1.2 \times 10^3$ cells ml$^{-1}$ for corrosion test and biofilm formation.

For bacteria adhesion assays, the culture medium was harvested by centrifugation for 10 min and resuspended in nutrient-rich artificial seawater to give a bacterial concentration of $7.38 \times 10^5 \pm 2.1 \times 10^4$ (assay 1) and $1.15 \times 10^6 \pm 1.6 \times 10^5$ (assay 2).

Additionally, 2.5% glutaraldehyde was prepared for fixing the bacterial biofilm for surface analyses scanning electron microscope with energy-dispersive X-ray analysis (SEM-EDX). Phosphate-buffered saline (PBS 1X), formaldehyde (4% in PBS) was prepared for staining bacteria with 4′,6-diamidino-2-phenylindole (DAPI) in order to count bacteria by fluorescence microscopy and observe biofilm formation by confocal laser microscopy (CLSM) (ZEISS LSM 510 META, Zeiss, Germany).

All experiments except the corrosion rate evaluation were carried out for 14 days at 37°C, which falls within the optimum temperature range for the growth of mesophilic bacteria. The experiment included corrosion testing (electrochemical testing, corrosion rate measurement and surface analysis) and adhesion testing. All tests except corrosion rate evaluation were performed in duplicate. The corrosion rate test was replicated three times.

**Table 1.** Chemical composition of test coupons.

| stainless steel type | chemical elements (%) | | | | | | | | |
|---|---|---|---|---|---|---|---|---|---|
| | Fe | Mn | S | V | Si | Cr | Ni | Cu | Mo |
| SS 410 | 84.767 | 0.622 | — | 0.068 | 0.81 | 13.429 | 0.209 | 0.038 | — |
| SS 420 | 87.013 | 0.444 | — | 0.12 | 1.172 | 11.206 | 0.045 | — | — |
| SS 316 | 67.954 | 1.695 | 0.114 | 0.078 | 0.577 | 16.622 | 10.691 | 0.399 | 1.87 |
| DSS 2205 | 66.318 | 1.678 | 0.053 | 0.108 | 0.432 | 22.1 | 6.116 | 0.304 | 2.891 |
| chromium | — | — | — | — | — | 99.99 | — | — | — |

**Table 2.** Surface tension parameters of water, formamide and glycerol [17].

| liquid | $\gamma^{total}$ | $\gamma^{AB}$ | $\gamma^{LW}$ | $\gamma^{+}$ | $\gamma^{-}$ |
|---|---|---|---|---|---|
| water | 72.8 | 51 | 21.8 | 25.5 | 25.5 |
| N, N-dimethylformamide (DMF) | 58 | 19 | 39 | 2.28 | 39.6 |
| glycerol | 64 | 30 | 34 | 3.92 | 57.4 |

## 2.3. Analytical methods

### 2.3.1. Contact angle measurement

The contact angle is the angle where a liquid–vapour interface meets a solid surface. The contact angles of materials were measured by using smartphone-based contact angle measurement instrument method [42] and this method has been proved to have an accuracy of 0.01% and matched the performance of a top traditional measurement instrument [43].

Two microlitres of high pure water, formamide and glycerol were used to measure the contact angle at room temperature. The test was repeated five times for each material and bacterial lawn with each liquid used and the average contact angles were determined. The images recorded were analysed by using ImageJ software.

The contact angle of bacteria was determined by using a bacterial lawn obtained by filtering 40 ml of culture medium through 0.45 µm filters. The filters were maintained for 30 min in Petri dishes containing 1% (w/v) agar with 10% (v/v) glycerol then finally fixed to glass slide by using double-sided tape [44].

The surface free energy of the liquids are presented in table 2 [45].

### 2.3.2. Hydrophobicity

The hydrophobicity of each material was determined through surface tension components of the materials. The surface tension of each material was calculated by the three-liquid method developed by van Oss [46] using the contact angle values obtained by high pure water, N, N-dimethyl formamide and glycerol.

The surface tension component of each material was calculated by the following equations:

$$(1 + \cos \theta_i) \times \gamma_{Li}^{tot} = 2 \times \left( \sqrt{\gamma_{Li}^{LW} \times \gamma_S^{LW}} + \sqrt{\gamma_{Li}^{+} \times \gamma_S^{+}} + \sqrt{\gamma_{Li}^{-} \times \gamma_S^{-}} \right) \quad (2.1)$$

$$\gamma^{tot} = \gamma^{LW} + 2\sqrt{\gamma^{+}\gamma^{-}}. \quad (2.2)$$

The hydrophobicity of each material was estimated by the following equation [47]:

$$\Delta G_{SWS}^{tot} = -2 \times \left( \sqrt{\gamma_S^{LW}} - \sqrt{\gamma_W^{LW}} \right)^2 4 \times \left( \sqrt{\gamma_S^{+} \times \gamma_W^{-}} + \sqrt{\gamma_S^{-} \times \gamma_W^{+}} - \sqrt{\gamma_S^{+} \times \gamma_S^{-}} - \sqrt{\gamma_W^{+} \times \gamma_W^{-}} \right), \quad (2.3)$$

where:

$\theta$ is the contact angle of each material in each liquid

$\gamma_L^{tot}$ is the total surface tension of each liquid

$\gamma^{LW}$ is Lifschitz–van der Waals (LW) (non-polar) component

$\gamma^+$ is electron acceptor (+) polar component

$\gamma^-$ is electron donor (−) polar component

$\Delta G_{SWS}^{tot}$ is the interfacial free energy of each material in the presence of water

The subscript $i$ is for each liquid used ($i$ = high-purity water, formamide and glycerol).

The terms $L$ and $S$ in the subscript represent liquid and solid, respectively.

The terms $S$ and $W$ in the subscript denote the solid and high-purity water.

It has been reported that if $\Delta G_{SWS}^{tot}$ is negative, the surfaces of solid samples have less affinity for water than the water molecules have for themselves, then they are hydrophobic [47]. If $\Delta G_{SWS}^{tot}$ is positive they are hydrophilic. The more negative $\Delta G_{SWS}^{tot}$ is, the more hydrophobic the surface and the more positive $\Delta G_{SWS}^{tot}$ is, the more hydrophilic is the surface.

### 2.3.3. Adhesion experiment, biofilm formation and sulfide concentration in the biofilm

To assess the early stage of bacterial attachment, assay 1 and assay 2 were distributed to Schott bottles with 400 ml each and each bottle containing one type of stainless steel coupon in triplicate. The bottles were incubated at 37°C. The coupons were fully immersed in the medium with the polished sides facing upward. After 2 h immersion, the coupons were gently rinsed three times with 1X PBS to remove loosely attached bacteria, then 350 µl of 4% formaldehyde was added to fix the cells for 20 min, rinsed three times with 1X PBS, stained with 350 µl of 300 nM DAPI in PBS solution and incubated without light for 5 min. The coupons were rinsed three times with 1X PBS before microscopic observation. Attached cells were enumerated by fluorescence microscopy and around 10–15 different fields were randomly selected and counted for each surface.

To observe biofilm formation and biogenic sulfide produced, two coupons of each type of stainless steel were immersed in media in a 500 ml glass bottle for 2 days. A set of coupons were also taken out, then the biofilm on the coupons was scraped off with a scalpel and dissolved in 10 ml of high pure water by vortex to create a suspension. The sulfide concentration of the suspension was measured using HACH DR 300 and sulfide reagents. The rest of the coupons were then taken out for observation under confocal laser scanning microscope (CLSM) ZEISS LSM 510 META for biofilm formation with the same preparation as samples for adhesion experiments.

The adhesion test was performed for two assays after 60 and 120 min immersion and after 2 days (CLSM results). The results after 60 min were similar to that after 120 min so this result was not shown. Hence, this work did not concentrate on bacterial adhesion rate, but it had been studied by other researchers [48].

### 2.3.4. Electrochemical testing

Electrochemical tests to measure corrosion resistance were performed on coupons following 14 days of immersion. This test was carried out in a three-electrode cell. A platinum-coated electrode was used as a counter electrode, an Ag/AgCl electrode was used as a reference electrode. A nitrogen gas layer was added to the top of the cell to create fully anaerobic conditions inside the cell. The electrochemical experiments were performed using VERSASTAT3-300 potentiostat and the results analysed using VersaStudio software. Open circuit potential (OCP) values of each specimen were recorded daily.

Electrochemical impedance spectroscopy (EIS) and potentiodynamic polarization were recorded at the end of the 14-day experiment. The EIS tests were carried out at OCP and the amplitude value was 10 mV with a frequency range from 0.05 to 100 000 Hz. The impedance data were analysed by an equivalent circuit using software ZsimpWin which was integrated with VersaStudio. The polarization curve was recorded potentiodynamically at a scan rate of 0.5 mV s$^{-1}$ starting from −0.25 V versus OCP to 1.5 V versus OCP. Corrosion potential and current density were obtained from the curve.

### 2.3.5. Corrosion rate and surface analysis

Coupons were weighed before immersing in the medium for six months at 37°C. At the end of the experiment, the coupons were taken out, washed three times with 100 ml high pure water, then cleaned in an ultrasonic bath for 2 min. The samples were then immersed in Clarke's solution according to ASTM standard G1-03 [49]. Finally, all samples were rinsed in high pure water followed

by 80% ethanol and dried in a biohazard cabinet. The cleaned samples were weighed and the surface analysed by SEM-EDX JEOL JXA-8200 EPMA WDS/EDS.

The corrosion rate (millimetre per year (mmpy)) of the coupons was measured according to ASTM standard G1-03 using weight loss measurements [49]:

$$\text{corrosion rate (mmpy)} = [8.76 \times 10^4 \times \text{weight loss (g)}]/[\text{density (g cm}^{-3}) \times \text{area (cm}^2) \times \text{time (h)}] \tag{2.4}$$

# 3. Results

## 3.1. Surface tension and hydrophobicity

Table 3 shows contact angle, surface tension components value and hydrophobicity of SS 410, SS 420, SS 316, DSS 2205, chromium and bacterial cells using three liquids including water, N, N-dimethylformamide and glycerol. Previous research indicated that if $\Delta G_{SWS}^{tot}$ is negative, the surfaces of solid samples have less affinity for liquid than the liquid molecules have for themselves, then they are hydrophobic [47] and vice versa. The more negative the value of $\Delta G_{SWS}^{tot}$, the more hydrophobic the material. All materials used for the experiment had negative surface free energy ($\Delta G_{SWS}^{tot} < 0$), thus they have hydrophobic surfaces. SS 410 had the lowest value of surface energy indicating it has the highest hydrophobicity, with DSS 2205 having the lowest hydrophobicity of the stainless steels tested. The surface free energy of bacterial cells was only slightly negative indicating that their cell surfaces are weakly hydrophobic.

## 3.2. Adhesion and biofilm formation

The attachment of bacteria to the surface of the different materials after 2 h immersion in different assays are shown in figure 1. The CLSM image is in reverse position i.e. the top plane refers to the materials surface. The thicknesses of the biofilms included in the manuscript were the maximum height of the biofilms and 3–4 photos per coupon of the biofilm were taken. The error values of the maximum heights of the biofilm are provided below. The highest density of cells was observed on SS 410 while the pure Cr sample had the lowest cell density after 2 h immersion. The dense biofilm formation on SS 410 was observed by CLSM after 2 days of immersion (figure 2) despite the low biofilm thickness of 54.3 ± 1.7 µm compared with SS 420 (68.4 ± 1.6 µm), SS 316 (68.1 ± 1.2 µm), DSS 2205 (57 ± 1.6 µm) and pure Cr coupon (64 ± 1.9 µm). DSS 2205 and SS 316 had high biofilm thickness yet the density of the biofilm was lower than SS 410. It is interesting to note that the distribution of the biofilm on chromium coupons was not uniform. Bacteria tends to form biofilm in clusters on the surface. This could suggest that the microorganisms attached to the favourable areas on the surface of materials.

## 3.3. Sulfide concentration in the biofilm formed on coupons' surfaces

For the first 2 days of immersion, the sulfide concentration of the biofilm suspension from SS 410 was approximately twice that of SS 316 and SS 2205 (figure 3). Pure chromium had the lowest biofilm sulfide concentration. This is in good agreement with the figure 2 where the biofilm formed on chromium coupons was less dense while the biofilm formed on SS 410 exhibited a highly dense film. The high concentration of sulfide in the biofilm where the pH was slightly acidic and the environment was anaerobic could accelerate the corrosion process in materials [21,50–52].

## 3.4. Electrochemical testing

Figure 4 shows OCP decay for all coupons, SS 410, SS 420, SS 316, DSS 2205 and pure chromium immersed in control solution (no SRB) and in the environment containing SRB for 14 days. The OCP values of coupons in the control solution remained relatively stable during the experiment. In the biotic solution, the OCP values of all coupons rapidly decreased over 2 days, followed by relatively stable values after 3 days. The test was conducted in stagnant conditions and the electrodes were placed face up. Therefore, along with the biofilm formation, there was also a film of bacteria including dead cells that had settled down on the surface of the materials. The thickness of the biofilm increased over time which made the transportation of ion through the biofilm more difficult and

**Table 3.** Surface tension components and hydrophobicity of the materials and bacterial cells and contact angle of SS 410, SS 420, SS 316, SS 2205, chromium and bacterial lawn with water, N, N-dimethylformanmide and glycerol.

| materials | | SS 410 | SS 420 | SS 316 | DSS 2205 | chromium | D. vulgaris |
|---|---|---|---|---|---|---|---|
| contact angle (°) | water | 83.12 ± 0.73 | 81.22 ± 2.38 | 69.10 ± 2.41 | 81.82 ± 3.02 | 58.57 ± 2.322 | 41.73 ± 1.75 |
| | N, N dimethylformamide | 46.53 ± 0.9 | 74.87 ± 4.5 | 77.03 ± 2.70 | 79.93 ± 4.37 | 53.40 ± 4.13 | 12.63 ± 1.69 |
| | glycerol | 39.03 ± 3.71 | 41.95 ± 3.35 | 33.97 ± 2.21 | 42.87 ± 1.13 | 21.8 ± 1.50 | 18.5 ± 0.65 |
| $\Delta G_{SWS}^{tot}$ (mJ m$^{-2}$) | | −17.86 | −5.93 | −9.61 | −5.65 | −5.97 | −0.77 |

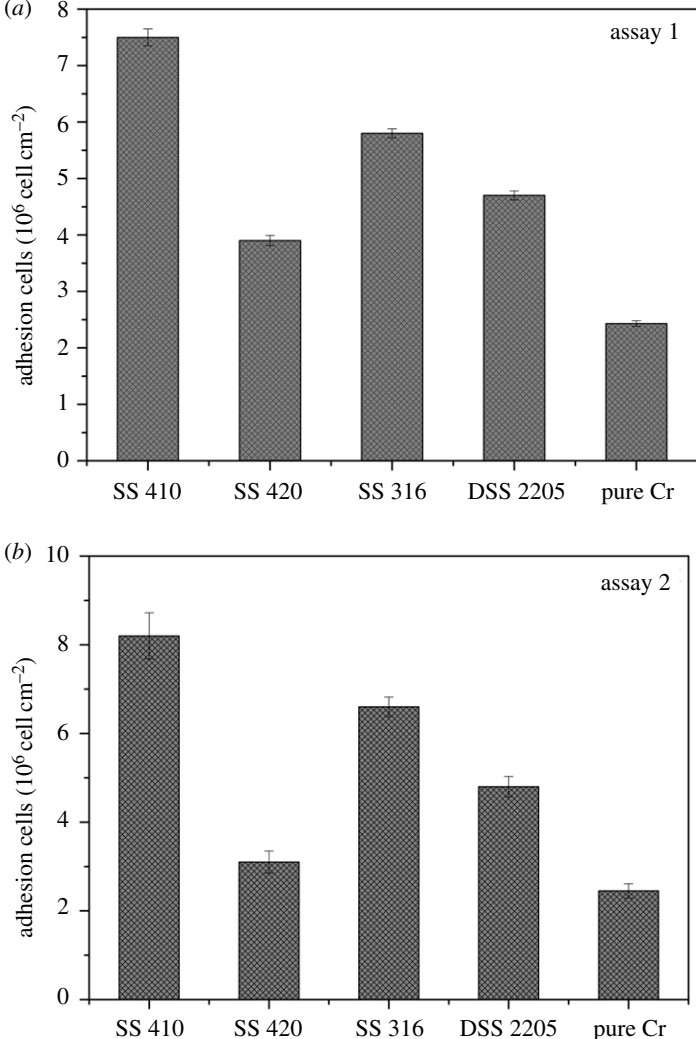

**Figure 1.** Density of cells adhering to SS 410, SS 420, SS 316, DSS 2205 and pure Cr samples using assay 1 (*a*) and assay 2 (*b*).

resulted in decreasing OCP values. This could explain why the OCP values of the coupons after a few days remained stable at low OCP values. The chromium sample OCP value remained more stable around −773 mV. The OCP values in all test stainless steel samples tended to increase after approximately one week. This suggests a strong susceptibility of stainless steel to corrosion. A previous study reported the local pH under the biofilm was slightly acidic [53], which is an important factor promoting pitting. Once corrosion occurred, the OCP started to increase. In comparison with other samples, the OCP value of SS 410 was higher than for other coupons and tended to increase during exposure, which reveals the strong corrosion reactions occurring after exposure. Other stainless steel samples with the same trend in OCP value suggest that all the test samples had a similar reaction in the MIC environment.

Figure 5 presents the EIS spectra (Nyquist plot) obtained at OCP of different materials after 14 days of exposure in both control solution (*a*) and biotic solution (*b*). The corresponding equivalent electric circuit (EEC) model for coupons immersed in control solution and in biotic solution are presented in the graphs. The Nyquist plots reveal a capacitive arc representing the resistance of the film formed on the electrode surface. The film can be a passive film, biofilm or even corrosion product layer. The higher the radius of the capacitive arc, the higher the resistance of the electrode to corrosion. Different EEC models have been proposed for interpreting impedance spectra of materials in the control environment and in the microbial environment. The EEC model $R_s[Q_{CPE}R_{ct}]$ has been used for control environment and the model $R_s[Q_{CPE}[R_b[C_{dl}R_{ct}]]]$ has been used for studying microbial corrosion [17,54]. A passive film along with biofilm formation on the surface of materials can act as a double-layer capacitance [17,54]. In this research, this model was used to fit the experimental data as it can represent the double-layer capacitance. $R_s$ is the resistance of solution, $R_b$ the resistance of biofilm/passive film formed on

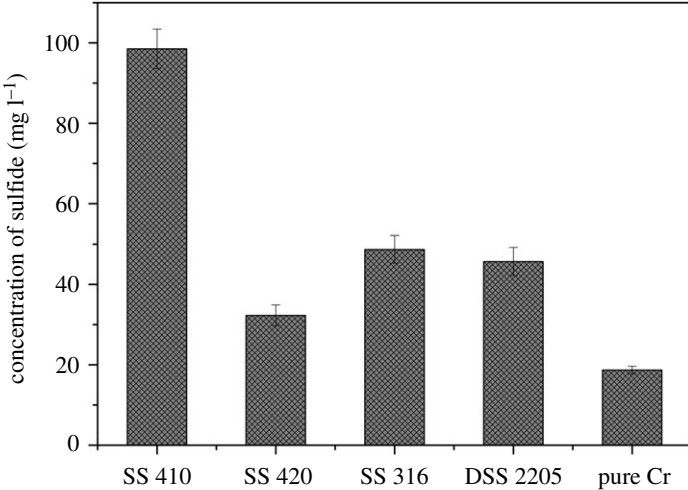

**Figure 2.** CLSM images of biofilm formation on tested coupons (*a*) SS 410, (*b*) SS 420, (*c*) SS 316, (*d*) DSS 2205 and (*e*) pure chromium.

**Figure 3.** Sulfide concentration presented in biofilm suspensions from coupons after 2 days.

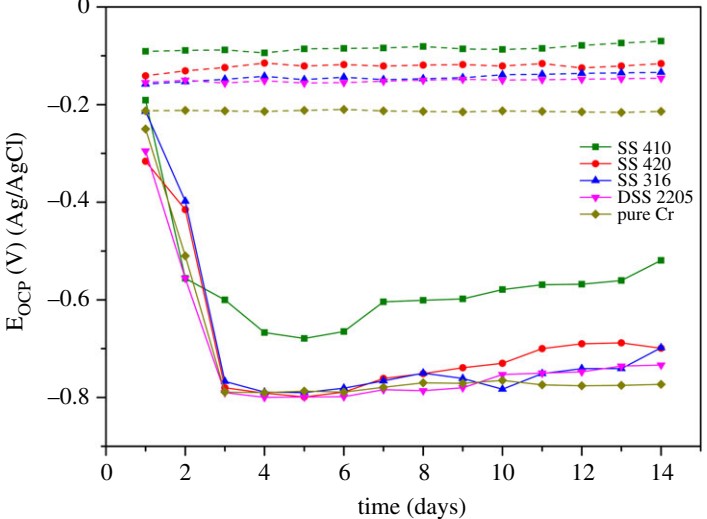

**Figure 4.** Open circuit potential (OCP) of each SS 410, SS 420, SS 316, DSS 2205 and pure chromium coupons during 14 days of exposure to SRB. The dashed and solid lines denote values in the absence and presence of SRB, respectively.

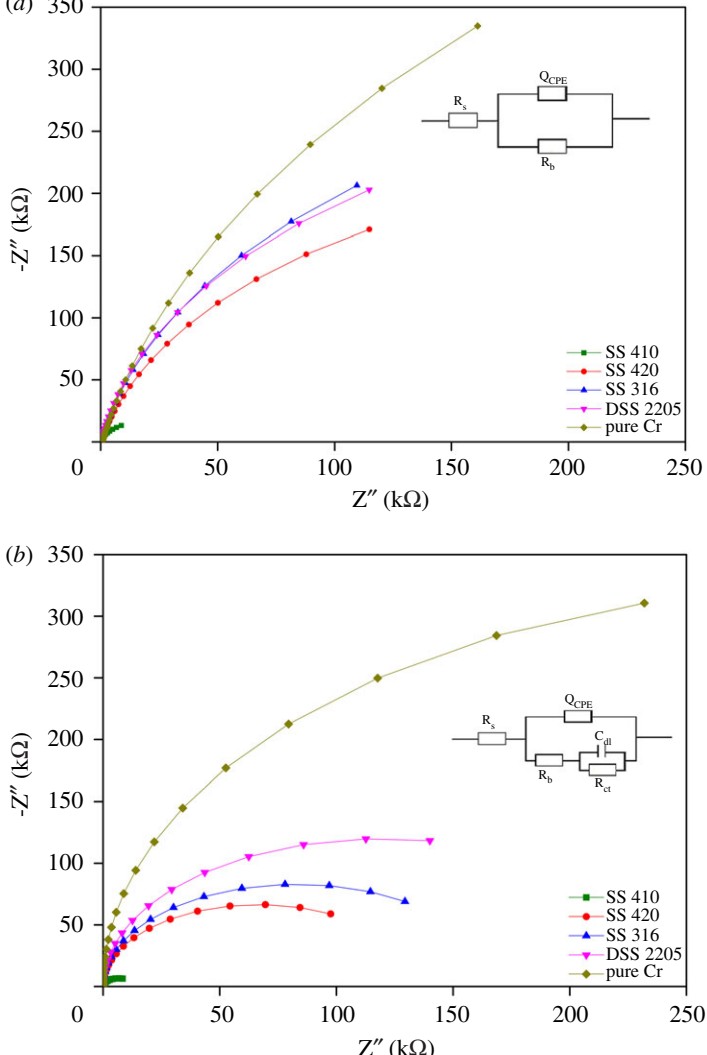

**Figure 5.** Nyquist plot of each stainless steel type obtained from the OCP values in control solution (*a*) and in biotic solution (*b*).

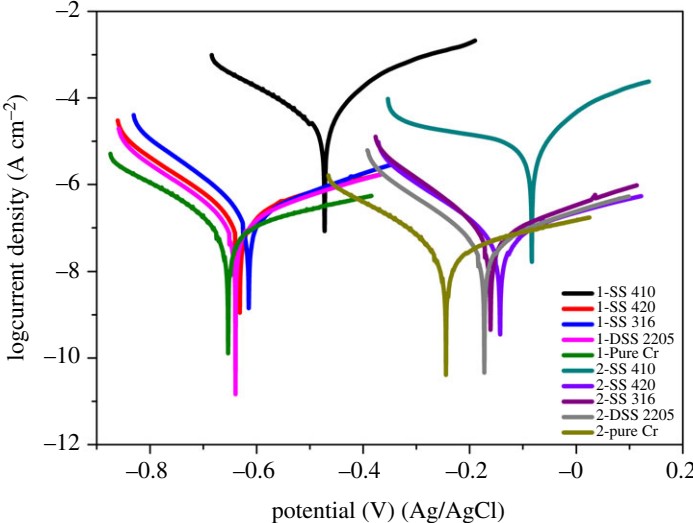

**Figure 6.** Polarization curves of tested coupons in control solution (1) and in bacteria environment (2).

**Table 4.** Corrosion potential and current density of all coupons in control and microbial solution.

| materials | | SS 410 | SS 420 | SS 316 | DSS 2205 | chromium |
|---|---|---|---|---|---|---|
| control solution | corrosion potential Ecorr (mV) (Ag/AgCl) | −83 | −143 | −161 | −173 | −244 |
| | current density Icorr (μA cm$^{-2}$) | 11.3 | 0.06 | 0.05 | 0.04 | 0.02 |
| microbial solution | corrosion potential Ecorr (mV) (Ag/AgCl) | −524 | −734 | −710 | −752 | −784 |
| | current density Icorr (μA cm$^{-2}$) | 56.6 | 0.19 | 0.14 | 0.12 | 0.06 |

material surfaces, $R_{ct}$ the charge transfer resistance and $C_{dl}$ the capacitance of electrical double layer. A constant phase element (CPE) was introduced to the model as it presents a deviation from a true capacitive behaviour. In general, the radius of capacitive arcs of all coupons immersed in control environment was higher than in the microbial environment. In the Nyquist plot, the radius of the capacitive arc defines the resistance of the protective film that forms on the surface of materials. The protective film can include the passive film of stainless steel and the biofilm of microorganism. The SS 420 sample exhibits the smallest capacitive radius while DSS 2205 shows the highest. This could mean that the protective film formed on DSS 2205 leads to better corrosion resistance than that formed on SS 410.

The polarization curves of all coupons obtained by potentiodynamic polarization test after 14 days of immersion in control solution and in microbial solution are presented in figure 6 and corrosion potential and current density are shown in table 4. As can be seen from figure 6, SS 410 polarization curves were in a higher position than for other materials, which indicates they are highly susceptible to corrosion in both control and biotic conditions. Current density represents the corrosion rate of materials. In general, the current density of all coupons immersed in microbial solution was higher than in the control condition, which suggests the acceleration of the corrosion process by the presence of SRB. SS 410 had the highest current density in both control and microbial solution which suggests poor corrosion resistance of SS 410.

## 3.5. Corrosion rates by weight loss

The corrosion rates of all coupons measured by weight loss after six months of immersion in both biotic and control conditions is presented in figure 7. Generally, the corrosion rate of tested coupons in the biotic condition was higher than in the control condition. SS 410 had corrosion rate in the biotic condition of around five times of that in the control condition while other tested coupons in the biotic condition had around three times of that in the control condition. It is in good agreement with the polarization data (table 4 and figure 6).

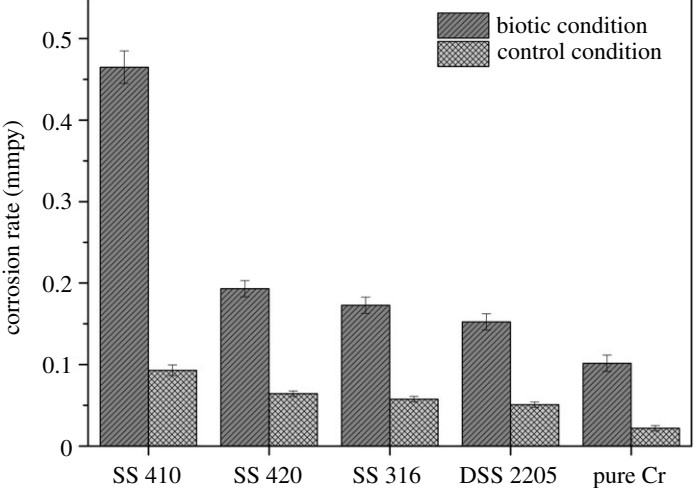

**Figure 7.** Corrosion rate of each coupon after six months of immersion.

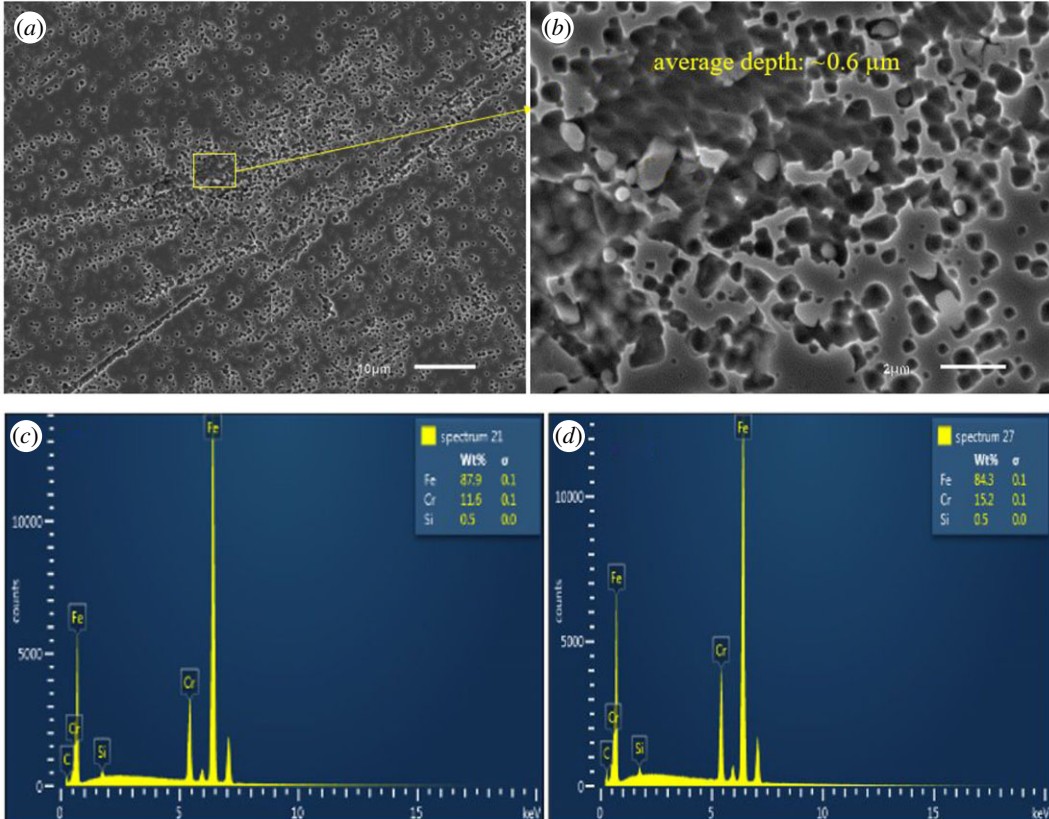

**Figure 8.** SEM-EDX image of SS 410 after six months of exposure in biotic condition.

In biotic condition, the corrosion rate of SS 410 was the highest compared with all other samples, approximately 4.5 times higher than for pure Cr. This could be due to the presence of higher sulfide and higher bacterial density in the biofilm (figures 1–3) formed on SS 410 than other tested materials. Chromium coupons exhibited the highest corrosion resistance.

## 3.6. Scanning electron microscope with energy-dispersive X-ray analysis

SEM-EDX images of all coupons immersed in biotic condition are shown in figures 8–12. The pure chromium sample had good corrosion resistance with very few pits formed on the surface.

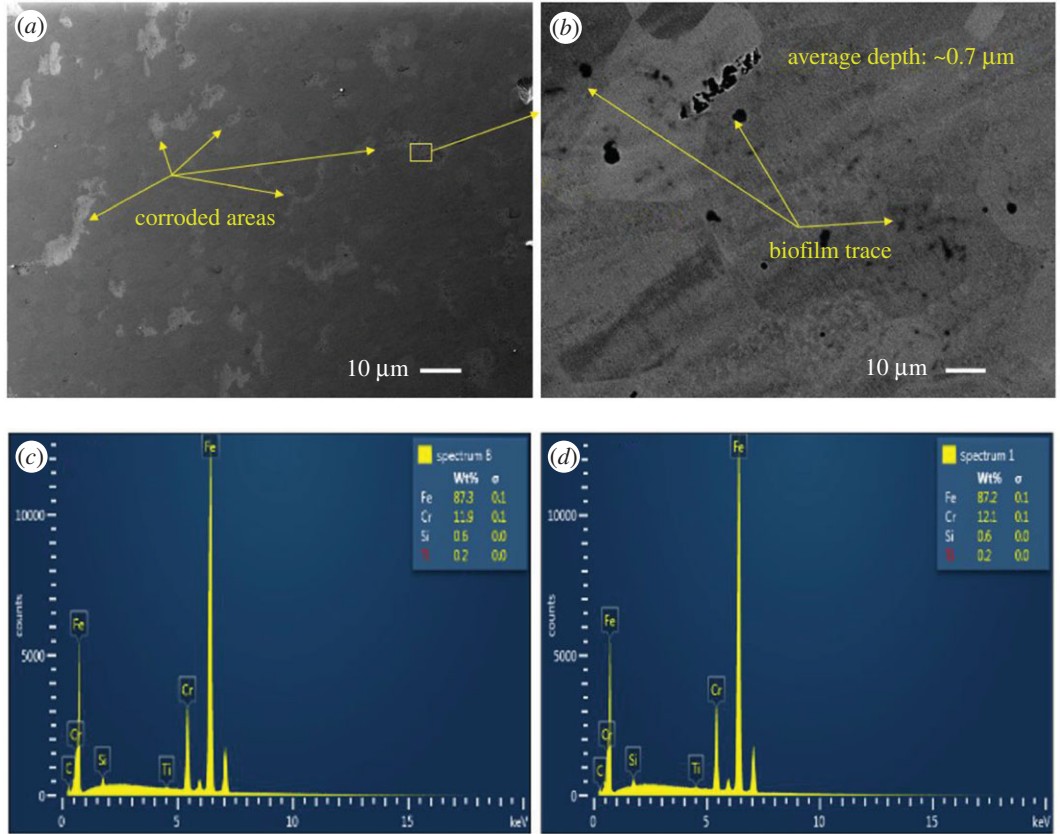

**Figure 9.** SEM-EDX image of SS 420 after six months of exposure in biotic condition.

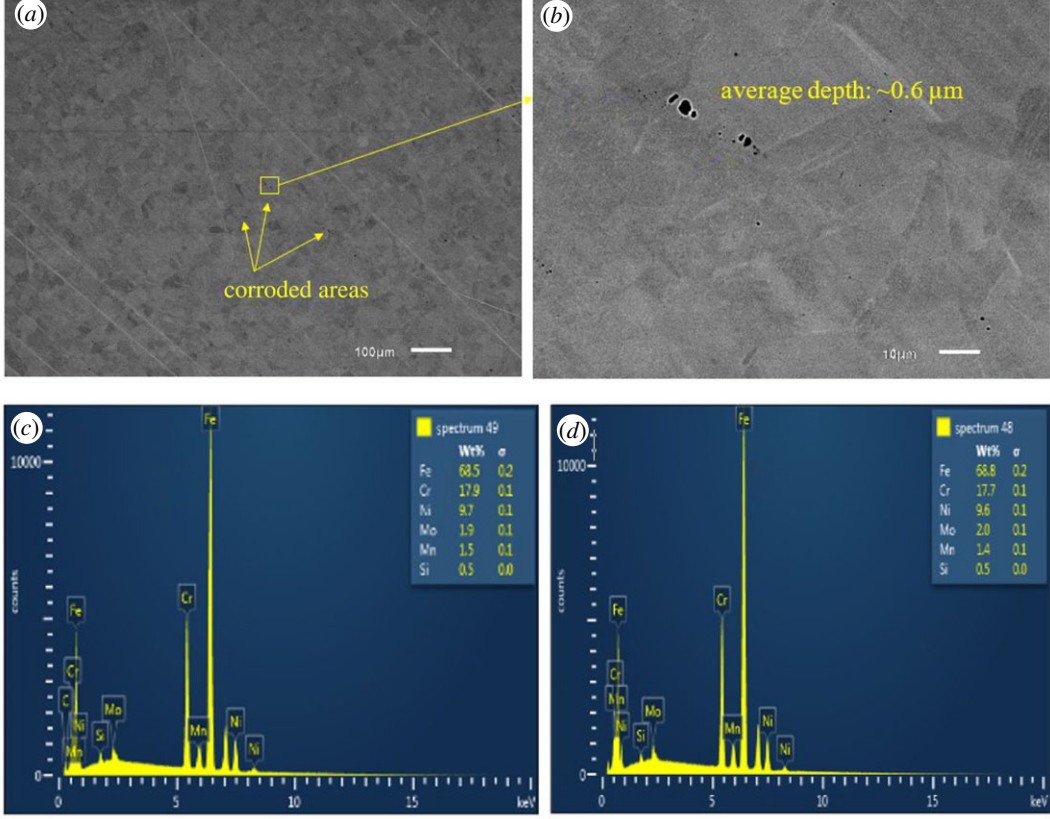

**Figure 10.** SEM-EDX image of SS 316 after six months of exposure in biotic condition.

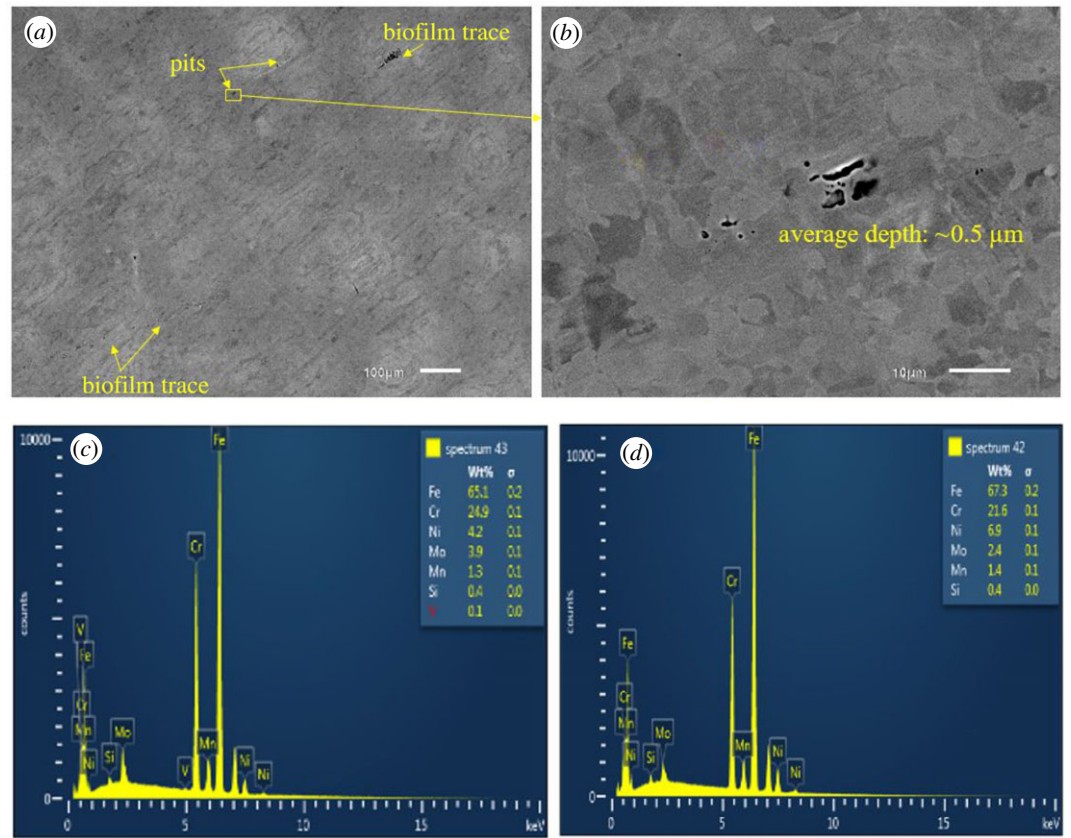

**Figure 11.** SEM-EDX image of DSS 2205 after six months of exposure in biotic condition.

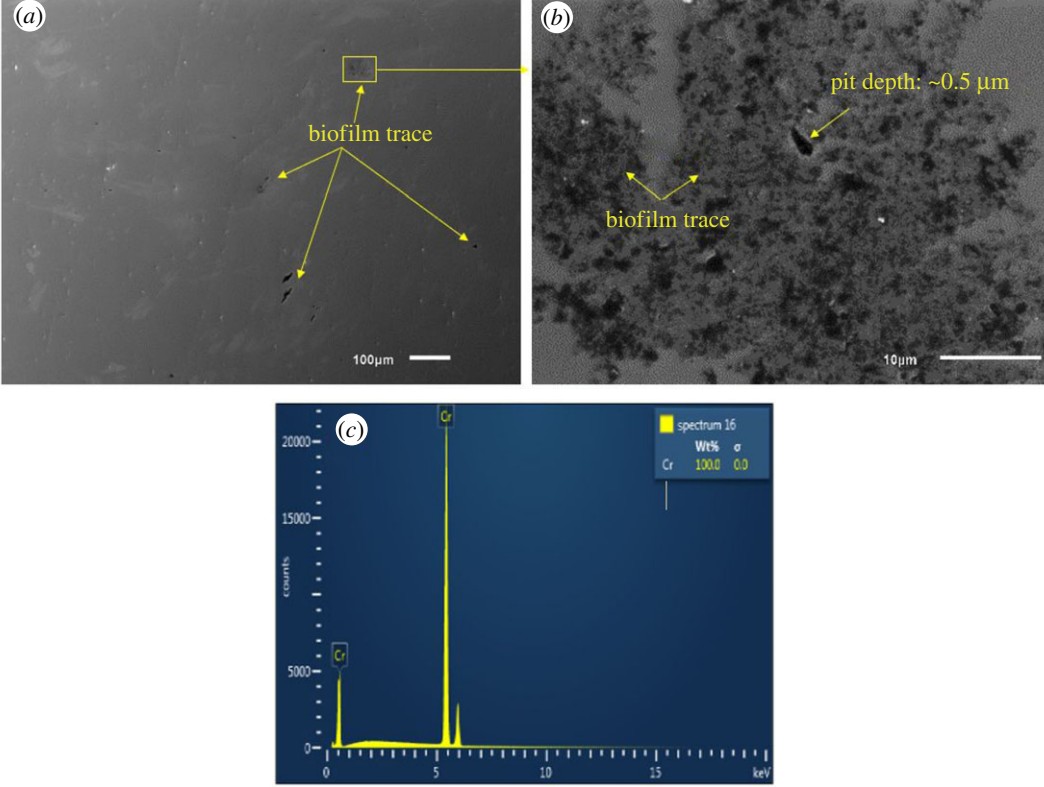

**Figure 12.** SEM-EDX image of pure chromium after six months of exposure in biotic condition.

With sample SS 410, region (*a*) shows pitting areas and region (*b*) shows non-pitting areas (figure 8). The EDS shows that non-pitting areas had higher chromium contents (15.2%) than pitting areas (11.5%) which suggests that corrosion happened in the depleted chromium area.

SS 420 is also martensitic stainless steel with slightly lower chromium content than SS 410 but had better corrosion resistance (figure 9).

There were less pits formed on the surface of SS 316 than SS 410 and SS 420 indicating better corrosion resistance in MIC environment (figure 10).

Figure 11 shows DSS 2205 also achieved very good corrosion resistance in MIC environment even though it had higher bacterial adhesion. The EDS spectra of DSS 2205 show that the lighter area (*a*) is austenite phase (higher nickel content and lower in molybdenum content) and the darker area (*b*) is ferrite phase. Most of the pits observed in the DSS 2205 coupons were found in the austenite phase. This can be explained by the higher nickel content in austenitic phase of DSS 2205 which can attract more bacteria to adhere to its surface [28] and is therefore more susceptible to MIC. Second, less chromium content in the austenite phase leads to less corrosion resistance than the ferrite phase. Therefore, most pits were formed in the austenite grains.

Only one pit was found on the chromium coupon which indicated its high corrosion resistance to MIC (figure 12).

Figure 13 shows SEM images of all coupons in the control condition. Generally, the total pits and the pit depth of each coupon are less than that in biotic condition. This confirms the corrosive ability of SRB to materials. SS 410 had the highest corrosion rate compared with other materials. In comparison with figure 8, the total number of pits and the size of the pits formed on SS 410 in the control condition were visually significantly lower than in biotic condition. This is also in agreement with figure 7 where the corrosion rate of SS 410 in biotic condition was nearly five times than in the control condition. Pure chromium exhibited insignificant pits.

# 4. Discussion

## 4.1. Adhesion of *Desulfovibrio vulgaris* on material surface

### 4.1.1. Hydrophobicity

The slightly negative value of the surface free energy of *D. vulgaris* shows it is weakly hydrophobic. This can be explained by evidence from studies with a Gram-negative *D. vulgaris* strain. This bacterium produces phospholipids, mainly phosphatidylethanolamine (PE) [55] which occur in biological membranes [56]. PE is synthesized by the addition of cytidine diphosphate-ethanolamine to diglycerides, releasing cytidine monophosphate. Several studies have considered the binding of PE to the outer cell membrane of bacteria [57,58]. The findings from these studies imply the hydrophilic end of PE binds to the outer cell surface and the hydrophobic moiety is directed into the environment. This would result in *D. vulgaris* exhibiting weak hydrophobicity. Previous studies show that the binding of PE produced by the microorganism affects the interaction between the bacterial cells and the substratum [59]. Since *D. vulgaris* has a surface layer with the hydrophobic part directed to the environment, the bacterial cells are attracted to hydrophobic substratum rather than hydrophilic substratum [59] because the cells will preferentially align in a way that the hydrophobic surfaces come into contact in order to reduce its exposure to water. Therefore, the correlation trend in bacterial attachment and hydrophobicity can be expressed as that the more hydrophobic the surface, the easier the bacterial cells–substratum interactions (See electronic supplementary material, figure S1). As can be seen from table 3, SS 410 had the highest hydrophobicity and resulted in the highest bacterial interaction, thus highest bacterial adhesion.

### 4.1.2. Impact of alloy elements in adhesion of *Desulfovibrio vulgaris*

SS 420, SS 316, DSS 2205 and chromium coupons had similar hydrophobicity as each other as they had similar surface free energy. However, SS 316 exhibited higher bacterial attachment. The possible explanation for this is the effect of the chemical composition of the materials on bacterial attachment. SS 316 had high nickel content and nickel content has been reported to increase bacterial attachment [14,28].

Chromium had the lowest bacterial adhesion compared with the other samples. In solution, chromium can release Cr (III) and/or Cr (VI). Cr (VI) is more toxic to microorganisms than Cr (III)

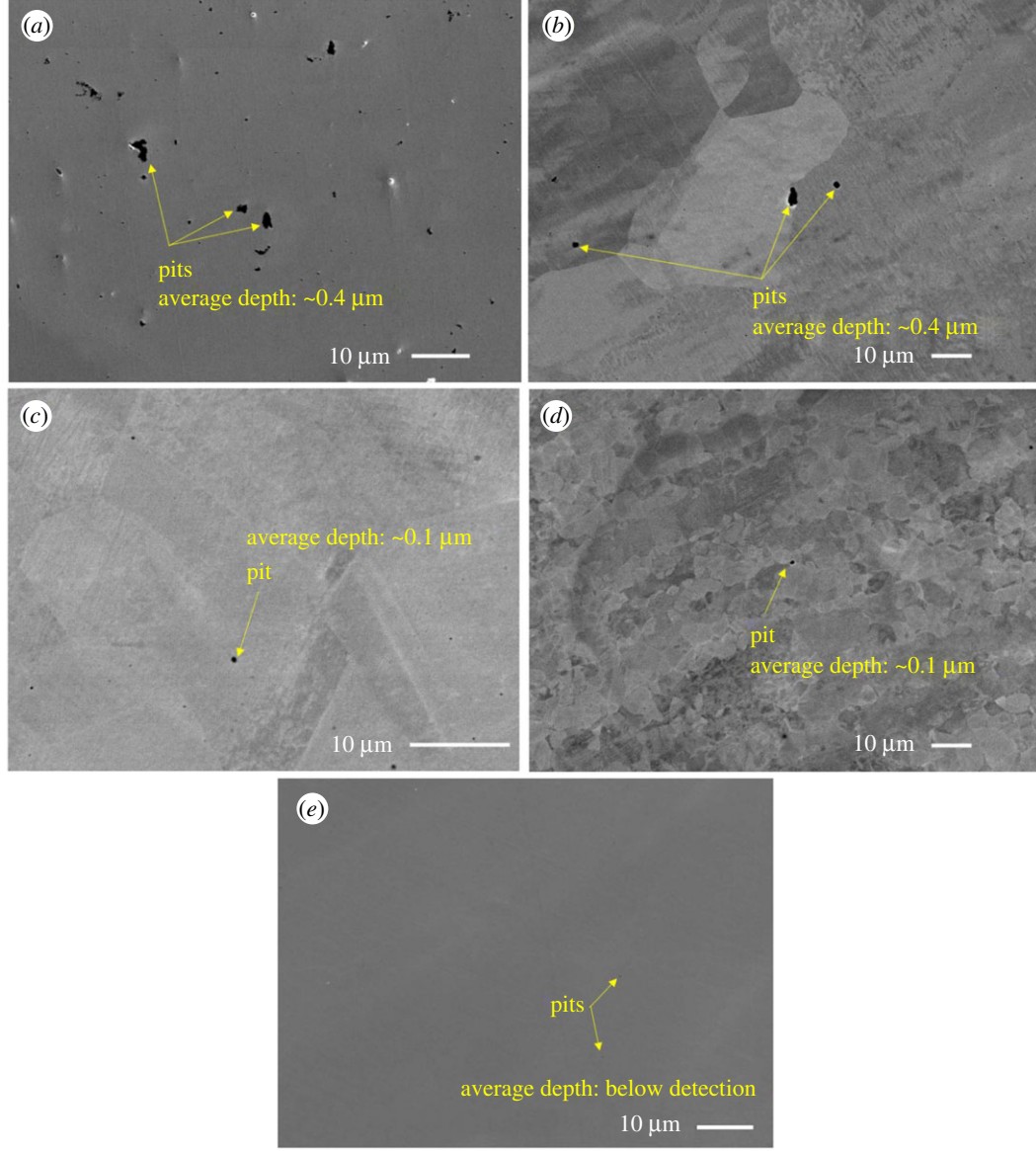

**Figure 13.** SEM images of tested coupons (*a*) SS 410, (*b*) SS 420, (*c*) SS 316, (*d*) DSS 2205 and (*e*) Pure chromium after six months of exposure in control condition.

and it was reported to be released from stainless steel in both *in vivo* and *in vitro* environments [34]. The release of Cr (III) from different stainless steel types have been reported in previous studies [24,25,60–62]. Fang *et al.* reported that Cr (III) was toxic to SRB even at very low concentrations [63]. This is most likely why the pure chromium sample in this study had the lowest bacterial attachment compared with other samples. Additionally, bacteria tended to form in clusters in narrow areas on the surface (figure 2*e*) which is different from the attachment of bacteria to the stainless steel coupons. This can be explained by the dissolution of chromium to Cr ion and increased the concentration of Cr ion on materials surface. SRB tends to form in a cluster in order to minimize the area exposed to toxic Cr ions [64].

### 4.1.3. Bacterial adhesion summary

Bacteria in the solution that come into contact with material surfaces can naturally form biofilms on the surface of materials. Bacterial attachment is considered as a precursor step to microbial corrosion and it can have a significant effect on the corrosion behaviour of materials at the material/solution interface [65,66]. The development of a biofilm may create various localized changes of the material surface and result in MIC. There are many factors that can influence bacterial adhesion on the surface of materials including physical and chemical composition of materials [14,28], physico-chemical

properties of microorganism [67] and environmental conditions [13]. Chemical compositions that have been reported to influence bacterial adhesion were mainly nickel, nitrogen and copper. Additionally, the alloying elements such as nickel, chromium and nitrogen added to steel could significantly affect the attachment of bacteria and biofilm development [22]. The hydrophobicity and chemical composition vary in each material considered in this study. Thus, the different behaviours observed with respect to bacterial adhesion are likely to be due to the effect of combination of these factors on the behaviour of the surface with respect to bacterial attachment. It appears in this study that chromium had a strong influence on bacterial attachment as it decreased the number of bacteria adhering to pure chromium coupons and the surface area to which bacteria adhered.

## 4.2. Corrosion behaviour of materials in artificial seawater containing *Desulfovibrio vulgaris*

### 4.2.1. The dissolution of the passive film

The formation of biofilm can lead to a local change in pH, dissolved oxygen, etc., which change the electrochemistry of the biofilm–metal system [68]. The pH within a biofilm tends to be slightly acidic [53], which could result in the dissolution of the passive film, which is mostly composed of ferrous oxide and chromium oxide due to the occurrence of the following reactions [69]:

$$Fe_2O_3 + 12H_2O + 6H^+ + 2e^- \rightarrow 2Fe(H_2O)_6^{2+} + 3H_2O \tag{4.1}$$

$$3Cr_2O_3 + 10H^+ \rightarrow 2Cr_3(OH)_4^{5+} + H_2O \tag{4.2}$$

$$Cr_2O_3 + H_2O + 2H^+ \rightarrow 2Cr(OH)_2^+. \tag{4.3}$$

Figure 14a illustrates schematically the dissolution of the passive film. The dissolution of the passive film leads to positively charged diffusion layers that attract $Cl^-$ and $S^{2-}$ ions produced by SRB metabolism which reach the surface due to electromigration [70,71]. This could lead to adsorption of these species in competition with -OH at the surface of the passive film and disrupt the passivation of stainless steel materials, thus leading to the breakdown of passivity.

### 4.2.2. Effect of bacteria in corroding materials

*D. vulgaris* corrode metals by electrochemical mechanisms through a series of oxidation (anodic) and reduction (cathodic) reactions of chemical species in direct contact with, or in close proximity to, the metallic surface. The mechanism of corrosion by SRB proposed by previous studies is schematically illustrated in figure 14b [72,73]. There are three ways in which electron uptake by SRB is facilitated, namely, through direct contact, conductive pilus (direct electron transfer (DET)) or electron mediator (mediated electron transfer (MET)) [72,73]. In DET, there is a need for direct contact between microorganisms and the steel surface, while MET involves soluble redox mediators, which are dependent on microbes [74,75].

The mechanism presented in figure 14 was based on the previous literature [72,73]. The breakdown of the passive film steps has been added.

The source of electrons can come from anodic reactions including iron ion and oxidation reaction of a carbon source from the environment. Carbon sources for SRB growth include hydrocarbons and fatty acids (e.g. formate, pyruvate, acetate, methanol and lactate) [6]. The anodic reactions can be expressed as follows [73,76]:

$$4Fe \rightarrow 4Fe^{2+} + 8e^- \tag{4.4}$$

$$CH_3CHOHCOO^- + H_2O \rightarrow CH_3COO^- + CO_2 + 4H^+ + 4e^-. \tag{4.5}$$

Cathodic reactions are the reduction of sulfate to sulfide caused by bacteria serving as biocathode and by redox mediators

$$SO_4^{2-} + 9H^+ + 8e^- \rightarrow HS^- + 4H_2O \tag{4.6}$$

$$Med_{red} + e^- \rightarrow Med_{ox}.$$

The corrosion products are created by the following reactions:

$$Fe^{2+} + S^{2-} \rightarrow FeS \tag{4.7}$$

$$Fe^{2+} + CO_2 + OH^- \rightarrow FeCO_3 + H^+. \tag{4.8}$$

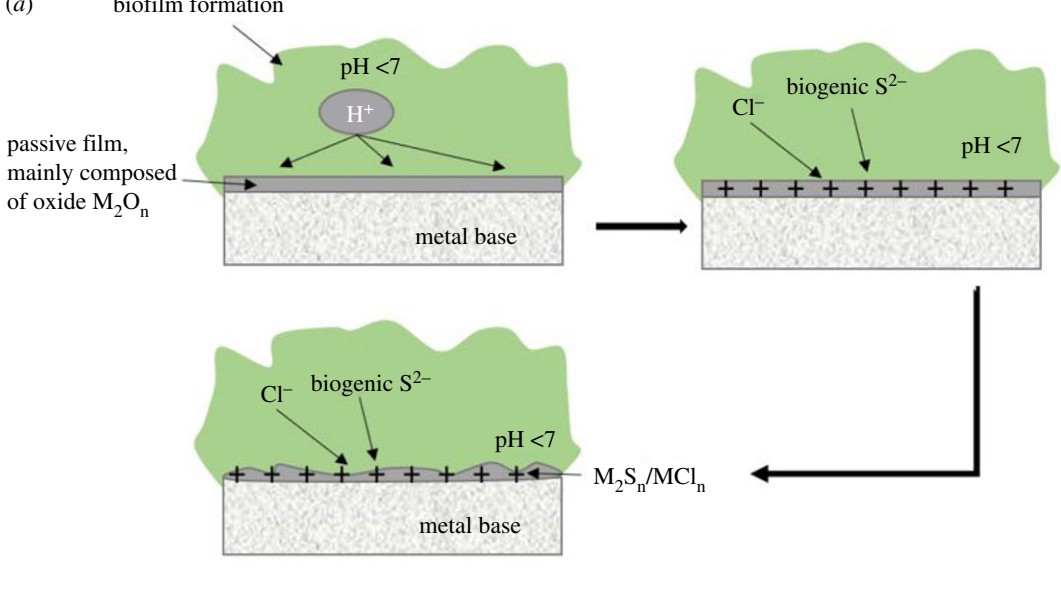

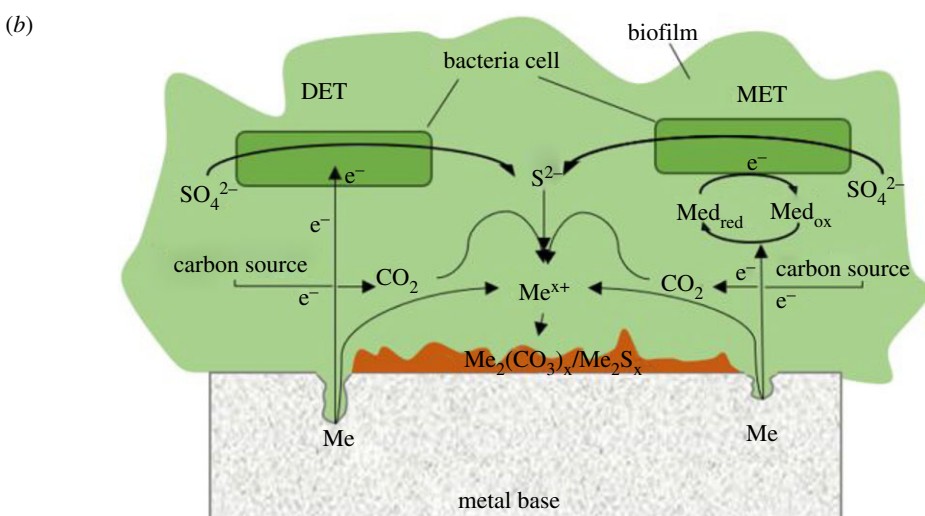

**Figure 14.** Corrosion mechanism of stainless steel under biofilm including breaking down of the passive film (*a*) and corrosion under biofilm (*b*).

Even though it is nutrient-rich medium, the electron donor is from both lactate and metal. The reactions can be written as follows:

$$4Fe + SO_4^{2-} + 8H^+ \rightarrow 4FeS + 4H_2O \tag{4.9}$$

$$2CH_3CHOHCOO^- + SO_4^{2-} + H^+ \rightarrow 2CH_3COO^- + 2CO_2 + HS^- + 2H_2O. \tag{4.10}$$

As per data from a previous study [76], Gibbs free energy of redox reaction (4.4) and (4.6) is $\Delta G° = -178$ kJ mol$^{-1}$ (at 25°C, pH 7) and of redox reaction (4.5) and (4.6) is $\Delta G° = -82.2$ kJ mol$^{-1}$. Thus, redox reaction (4.4) and (4.6) is thermodynamically more favourable than redox reaction (4.5) and (4.6). In other words, iron oxidation is more energetically favourable than lactate oxidation. Therefore, in a solution that contains both lactate and iron, electron donor could come from both lactate and iron.

The combined effects of bacteria metabolism that consumes electrons and metabolic products which are mainly sulfide ions make most metallic materials susceptible to sulfate-reducing bacteria-induced corrosion. The current density of SS 410 exposed to biotic environment was about five times that in the control environment, which could be explained by these effects.

SS 410 had the highest density of attached bacteria and the highest sulfide levels within the biofilm; thus, it was most susceptible to MIC compared with other samples. This can be compared between SS

410 and SS 420 which are both martensitic stainless steels. SS 410 had 2% chromium higher than SS 420. However, the Ni content of SS 410 was 0.2% compared with 0.045 for SS420. It appears that Ni seems to have a stronger effect in reducing hydrophobicity value and attracted more bacteria. However, in the case of SS 316 although the Ni content was about 10.7%, the MIC resistance was greater due to the presence of Mo in stabilizing the passive film even though it attracted more bacteria. Also, SS 410 had the densest biofilm formation (figure 2a), thus the diffusion of lactate to the biofilm was more limited than others. In the condition of lacking nutrients, e.g. lactate as carbon source, the electron donor comes from metal [76], thus this resulted in severe pitting formation on the surface of SS 410.

### 4.2.3. Impact of alloy elements in biocorrosion of stainless steel by *Desulfovibrio vulgaris*

Alloying elements are added to stainless steel to strengthen its mechanical properties and improve corrosion resistance not only in control but also in biotic environments. The passive film layer of stainless steel has a complex structure and complex chemical composition. It is normally described as a mixed metal oxide/hydroxide. Not only is the enrichment of chromium in the passive film a key factor for corrosion resistance [77], but the presence of molybdenum in the passive film can also improve corrosion resistance. The latter is reported to be able to mitigate the breakdown of the passive film or promote passive film repair after breakdown [78]. Previous work has also shown the formation of $MoO_2$ in the passive film helps the film to maintain its passivity at lower pH values [23]. This study showed that SS 316 and DSS 2205 had greater bacterial adhesion than SS 420, but a lower corrosion rate. This can be explained by the protection of the passive film layer of the samples. The higher corrosion resistance of SS 316 and DSS 2205 may be attributed to the more stable passive film due to the presence of $MoO_2$. Previous studies also suggest the high protection of stainless steel to MIC by molybdenum in the passive film [28] in the presence of sulfide ions in the environment. The role of molybdenum in pitting resistance can include: highly stable formation of molybdenum oxide in a slightly acidic environment; Mo can slow the dissolution kinetics after pitting initiation [79]; the presence of Mo in the surface of materials can adsorb sulfide species and desorb it [80], thus improving corrosion resistance when sulfide species are present. This can be seen in the Nyquist plot (figure 5) where SS 316 and DSS 2205 had higher resistance than SS 420 and SS 410. Furthermore, the current density of SS 316 and DSS 2205 in the biotic environment was around 2.8 times higher than in the control environment while SS 410 had around 5 times higher and SS 420 had around 3.1 times higher current density in biotic versus control environments. This clearly shows the high corrosion protection ability of chromium and molybdenum in an MIC environment.

Additionally, the pure chromium sample had the lowest corrosion rate and the fewest pits formed on the surface. The whole surface of the pure chromium sample was covered with chromium and there might be the presence of $Cr_2O_3$ which enhanced high corrosion resistance. The releasing of chromium ions on the surface by the pure chromium coupons results in less bacterial attachment than stainless steel samples as chromium ion was toxic to bacteria community. The biofilm formation in clusters on the pure chromium surface decreased the exposure of bacteria to chromium ion. This resulted in decreasing the presence of biogenic sulfur produced by bacterial metabolism in the biofilm matrix.

MIC is affected by a variety of factors and in this paper, the focus was on the combined effect of hydrophobicity, stability of the passive film by alloying elements and the toxicity of chromium on the resulting MIC. Because of the complex interaction of multiple factors, it is not a straightforward problem. Overall, the correlation between cell adhesion on the surface of materials and corrosion rate was weak (see electronic supplementary material, figure S2). In other words, the extent of bacterial attachment on the surface of test materials does not have a significant effect on the corrosion rate of materials. The corrosion resistance of stainless steel was dependent on the combined effect of all the alloying elements added to the stainless steel which strengthen the passivity of the passive film. The presence of molybdenum in the passive film also plays an important role in corrosion resistance in a microbial environment. Chromium also showed a significant influence in reducing bacterial attachment and consequent corrosion behaviour as pure chromium had the lowest bacterial attachment.

Some studies suggest to make stainless surfaces more hydrophobic to avoid corrosion [81,82]. However, in the microbial environment, hydrophobic bacteria might come into contact with hydrophobic surfaces and cause corrosion. Thus, using the hydrophobic surface to control corrosion method should be reconsidered in the microbial environment.

This study suggests that stainless steel with a chromium coating could improve corrosion resistance to MIC. However, the toxicity of chromium ions in the environment and the high cost of chromium coating should be taken into account. More studies are needed to ascertain this.

# 5. Conclusion

In this study, the mechanism of bacterial adhesion and corrosion behaviour of stainless steel was studied.

The hydrophobic material surfaces attracted *D. vulgaris*. Nickel appears to have an ability to attract bacteria to adhere on the surface.

The corrosion rate of different materials is determined not only by bacterial attachment but also stability of the passive film which is determined by the alloying elements, such as Mo and Cr. This can be clearly seen in SS 316, which had high bacterial attachment but lower corrosion rate than SS 420 as it has a more stable passive film with $MoO_2$ formation.

Pure Cr showed higher resistance to corrosion under MIC condition due to its toxicity to bacteria.

Data accessibility. Data available from the Dryad Digital Repository: https://doi.org/10.5061/dryad.18931zcvg [83].

Authors' contributions. T.T. conceived, designed the work, analysed data and wrote the manuscript. K.K., A.P. and S.T. assisted in designing the work, analysing data and revised the manuscript. All authors read and approved the final manuscript to be published.

Competing interests. We declare we have no competing interests.

Funding. This research was supported by an Australian Research Training Program Scholarship (grant no. 302787) provided through Charles Darwin University.

Acknowledgements. The authors acknowledge the support of Charles Darwin University technical staff in preparing experiment; The Centre for Microscopy and Microanalysis Centre in University of Queensland for doing SEM and EDX and Dr. Steven Mason, School of Chemistry and Molecular Biosciences, The University of Queensland for imaging with ZEISS LSM 510 META confocal laser scanning microscope.

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
