## [Reviewer comments · Royal Society Open Science]

Review History

RSOS-201577.R0 (Original submission)

Review form: Reviewer 1

Is the manuscript scientifically sound in its present form?

Yes

Are the interpretations and conclusions justified by the results?

Yes

Is the language acceptable?

Yes

Do you have any ethical concerns with this paper?

No

Have you any concerns about statistical analyses in this paper?

No

Recommendation?

Major revision is needed (please make suggestions in comments)

Comments to the Author(s)

This manuscript represents about adhesion of SRB and corrosion of some stainless steels. Authors claimed the correlation between hydrophobicity of metal surface and bacterial adhesion and higher corrosion resistant of pure chromium against MIC. However, the focus is divergent and the main message is difficult to understand. Especially, corrosion resistant in pure chromium is not due to alloy composition and toxicity of chromium ion for microbe probably contribute to corrosion resistant; namely, it is out of focus. In addition, as authors described in the manuscript, "The corrosion behavior of materials in MIC did not depend mostly on bacteria attachment but also on stainless steel type and composition". Therefore, I couldn't understand what the author wants to claim in this paper. Authors should clearly claim their opinion.

Major comments

1. Authors should clearly describe about their conclusion in the Summary. They described about importance of hydrophobicity in the bacterial adhesion, but the bacterial adhesion does not influence the corrosion rate. What is important things in this paper?
2. Authors described about the difference of adhesion cells in different materials. I think that adhesion test should be conducted for different immersion time. For example, Wakai and Harayama clearly showed the adhesion speed of each bacteria in the paper (Materials Technology, 2015, 30.sup5: B38-B43. Doi: /10.1179/1753555714Y.0000000220).
3. Fig. 2: How to determine the top and bottom of biofilm? Authors described about thickness of biofilm without any error values. Are these values from single view? In addition, biofilm in the panel (b) is most ununiformed. If it viewing is correct, adhesion bacterial cells on the surface of SS420 is lowest. Author should show reliable thickness of biofilm with error values based on some observation or viewing.
4. In SEM-EDS analysis, abiotic controls are missing. Especially, observation of SS410 in abiotic condition is significantly important.
5. Fig. 13: I cannot accept this corrosion mechanism. In this paper, authors did not show any evidence supporting this mechanism. Breaking down of passive film is important in the corrosion of stainless steel and this model does not include information of important event. Furthermore, since authors used nutrient rich medium in the corrosion experiment, electron donor for SRB is lactate does not metal. Most recently, Deng et al. reported important of biogenic iron sulfide nanoparticle in MIC by *D. vulgaris*. Please update the proposal mechanism.
6. SS410 and SS420 are both martensitic stainless steel and chromium content of SS420 is lesser than that of SS410. However, the corrosion of SS410 was sever. Furthermore, the corrosion of austenitic stainless steel SS316 proceeded in the less chromium austenite phase. Could you please discuss it more?

Minor Comments

1. How to measure hydrophobicity of bacterial cells. Although method of stainless steel was written, those of bacterial cells is missing.
2. Don't use rpm without rotor radius. Author should describe as $\times g$ or rpm with rotor radius.
3. Fig. 1: Both panels are "Assay 1". Please change the bottom panel to be "Assay 2".

Review form: Reviewer 2

Is the manuscript scientifically sound in its present form?

Yes

Are the interpretations and conclusions justified by the results?

Yes

Is the language acceptable?

Yes

Do you have any ethical concerns with this paper?

No

Have you any concerns about statistical analyses in this paper?

No

Recommendation?

Accept with minor revision (please list in comments)

Comments to the Author(s)

The author in the research paper "A study of bacteria adhesion and microbial corrosion on different stainless steels in environment containing *Desulfovibrio vulgaris*" they explain to detail the protection of corrosion in different steels using a Gram-negative bacteria.

I enjoyed to read the paper. The results and discussion correspond to the methods and introduction. I have a few comments.

P3L29-32. The author think that the bacterial cell-wall and membrane have an influence in the surface attachment.

P3L32-34. *D. vulgaris* is an aerobic strain... it can growth in low oxygen conditions?

P4L23. Could you add a reference?

P4L25-26. Why the authors chose those concentrations?

General

The authors need to homogenise the term hours or hr, using h. Also, N,N-dimethylformamide reagent.

Regarding to adherence. Do the authors know if exist reports about of the gene expression for adhesion to different materials?

Decision letter (RSOS-201577.R0)

Dear Dr Tran Thi Thuy:

Title: A study of bacteria adhesion and microbial corrosion on different stainless steels in environment containing *Desulfovibrio vulgaris*

Manuscript ID: RSOS-201577

The editor assigned to your manuscript has now received comments from reviewers. We would like you to revise your paper in accordance with the referee and Subject Editor suggestions which

can be found below (not including confidential reports to the Editor). Please note this decision does not guarantee eventual acceptance.

Please submit your revised paper before 11-Nov-2020. Please note that the revision deadline will expire at 00.00am on this date. If we do not hear from you within this time then it will be assumed that the paper has been withdrawn. In exceptional circumstances, extensions may be possible if agreed with the Editorial Office in advance. We do not allow multiple rounds of revision so we urge you to make every effort to fully address all of the comments at this stage. If deemed necessary by the Editors, your manuscript will be sent back to one or more of the original reviewers for assessment. If the original reviewers are not available we may invite new reviewers.

On behalf of the Subject Editor Professor Anthony Stace and the Associate Editor Dr Dattatray Late.

RSC Associate Editor:
Comments to the Author:
(There are no comments.)

RSC Subject Editor:
Comments to the Author:
(There are no comments.)

Reviewers' Comments to Author:

Reviewer: 1

Comments to the Author(s)

This manuscript represents about adhesion of SRB and corrosion of some stainless steels. Authors claimed the correlation between hydrophobicity of metal surface and bacterial adhesion and higher corrosion resistant of pure chromium against MIC. However, the focus is divergent and the main message is difficult to understand. Especially, corrosion resistant in pure chromium is not due to alloy composition and toxicity of chromium ion for microbe probably contribute to corrosion resistant; namely, it is out of focus. In addition, as authors described in the manuscript, "The corrosion behavior of materials in MIC did not depend mostly on bacteria attachment but also on stainless steel type and composition". Therefore, I couldn't understand what the author wants to claim in this paper. Authors should clearly claim their opinion.

Major comments

1. Authors should clearly describe about their conclusion in the Summary. They described about importance of hydrophobicity in the bacterial adhesion, but the bacterial adhesion does not influence the corrosion rate. What is important things in this paper?
2. Authors described about the difference of adhesion cells in different materials. I think that adhesion test should be conducted for different immersion time. For example, Wakai and Harayama clearly showed the adhesion speed of each bacteria in the paper (Materials Technology, 2015, 30.sup5: B38-B43. Doi: /10.1179/1753555714Y.0000000220).
3. Fig. 2: How to determine the top and bottom of biofilm? Authors described about thickness of biofilm without any error values. Are these values from single view? In addition, biofilm in the panel (b) is most ununiformed. If it viewing is correct, adhesion bacterial cells on the surface of SS420 is lowest. Author should show reliable thickness of biofilm with error values based on some observation or viewing.
4. In SEM-EDS analysis, abiotic controls are missing. Especially, observation of SS410 in abiotic condition is significantly important.
5. Fig. 13: I cannot accept this corrosion mechanism. In this paper, authors did not show any evidence supporting this mechanism. Breaking down of passive film is important in the corrosion of stainless steel and this model does not include information of important event. Furthermore, since authors used nutrient rich medium in the corrosion experiment, electron donor for SRB is lactate does not metal. Most recently, Deng et al. reported important of biogenic iron sulfide nanoparticle in MIC by *D. vulgaris*. Please update the proposal mechanism.
6. SS410 and SS420 are both martensitic stainless steel and chromium content of SS420 is lesser than that of SS410. However, the corrosion of SS410 was sever. Furthermore, the corrosion of austenitic stainless steel SS316 proceeded in the less chromium austenite phase. Could you please discuss it more?

Minor Comments

1. How to measure hydrophobicity of bacterial cells. Although method of stainless steel was written, those of bacterial cells is missing.
2. Don't use rpm without rotor radius. Author should describe as $\times g$ or rpm with rotor radius.
3. Fig. 1: Both panels are "Assay 1". Please change the bottom panel to be "Assay 2".

Reviewer: 2

Comments to the Author(s)

The author in the research paper "A study of bacteria adhesion and microbial corrosion on different stainless steels in environment containing *Desulfovibrio vulgaris*" they explain to detail the protection of corrosion in different steels using a Gram-negative bacteria.

I enjoyed to read the paper. The results and discussion correspond to the methods and introduction. I have a few comments.

P3L29-32. The author think that the bacterial cell-wall and membrane have an influence in the surface attachment.

P3L32-34. *D. vulgaris* is an aerobic strain... it can growth in low oxygen conditions?

P4L23. Could you add a reference?

P4L25-26. Why the authors chose those concentrations?

General

The authors need to homogenise the term hours or hr, using h. Also, N,N-dimethylformamide reagent.

Regarding to adherence. Do the authors know if exist reports about of the gene expression for adhesion to different materials?

Author's Response to Decision Letter for (RSOS-201577.R0)

See Appendix A.

RSOS-201577.R1 (Revision)

Review form: Reviewer 1

Is the manuscript scientifically sound in its present form?

Yes

Are the interpretations and conclusions justified by the results?

Yes

Is the language acceptable?

Yes

Do you have any ethical concerns with this paper?

No

Have you any concerns about statistical analyses in this paper?

No

Recommendation?

Accept as is

Comments to the Author(s)

This manuscript is carefully and conscientiously revised, and clearly resolved the problems.

Review form: Reviewer 2

Is the manuscript scientifically sound in its present form?

Yes

Are the interpretations and conclusions justified by the results?

Yes

Is the language acceptable?

Yes

Do you have any ethical concerns with this paper?

No

Have you any concerns about statistical analyses in this paper?

No

Recommendation?

Accept as is

Comments to the Author(s)

The authors responded to the comments and the paper has improvements in the methods, results and discussion

Decision letter (RSOS-201577.R1)

Dear Dr Tran Thi Thuy:

Title: A study of bacteria adhesion and microbial corrosion on different stainless steels in environment containing *Desulfovibrio vulgaris*
Manuscript ID: RSOS-201577.R1

It is a pleasure to accept your manuscript in its current form for publication in Royal Society Open Science. The chemistry content of Royal Society Open Science is published in collaboration with the Royal Society of Chemistry.

On behalf of the Subject Editor Professor Anthony Stace and the Associate Editor Dr Dattatray Late.

RSC Associate Editor:
Comments to the Author:
Accept as is

RSC Subject Editor:
Comments to the Author:
(There are no comments.)

Reviewer(s)' Comments to Author:
Reviewer: 2

Comments to the Author(s)
The authors responded to the comments and the paper has improvements in the methods, results and discussion

Reviewer: 1

Comments to the Author(s)
This manuscript is carefully and conscientiously revised, and clearly resolved the problems.

Appendix A

Dear reviewers,

Thank you for giving us the opportunity to submit a revised draft of our manuscript. We appreciate the time and effort that you have dedicated to providing the valuable feedback on our manuscript. We are grateful to you for your insightful comments on our paper. We have been able to incorporate changes to reflect the suggestions provided by the reviewers. We have highlighted the changes within the manuscript.

Reviewer: 1

Comments to the Author(s)

This manuscript represents about adhesion of SRB and corrosion of some stainless steels. Authors claimed the correlation between hydrophobicity of metal surface and bacterial adhesion and higher corrosion resistant of pure chromium against MIC. However, the focus is divergent and the main message is difficult to understand. Especially, corrosion resistant in pure chromium is not due to alloy composition and toxicity of chromium ion for microbe probably contribute to corrosion resistant; namely, it is out of focus. In addition, as authors described in the manuscript, "The corrosion behavior of materials in MIC did not depend mostly on bacteria attachment but also on stainless steel type and composition". Therefore, I couldn't understand what the author wants to claim in this paper. Authors should clearly claim their opinion.

Response:

MIC is affected by a variety of factors and in this paper, we have focussed on the combined effect of hydrophobicity, stability of the passive film by alloy composition, the toxicity of chromium on the resulting MIC. Because of the complex interaction of multiple factors, it is not straight forward problems. And this is addressed in the following responses below and we have made changes to the manuscript to make the focus clearer to the reader.

Major comments

1. Authors should clearly describe about their conclusion in the Summary. They described about importance of hydrophobicity in the bacterial adhesion, but the bacterial adhesion does not influence the corrosion rate. What are important things in this paper?

Response:

The important things include:

- Hydrophobicity had strong effect on bacterial attachment. Alloy elements e.g. nickel also had shown its ability to attract bacteria to adhere on the surface.
- However, corrosion rate of different materials is determined not only by bacterial attachment but also stability of the passive film which is determined by the alloying elements, such as Mo, Cr, etc. This can be clearly seen in SS 316, it had high bacterial attachment but lower corrosion rate than SS 420 as it has more stable passive film with MoO₂ formation
- Chromium was included in the manuscript is in order to show its toxicity on bacterial attachment and corrosion resistance. This is to investigate the value of chromium coating and preventing MIC.

We have modified the conclusion and abstract to to reflect the response above.

2. Authors described about the difference of adhesion cells in different materials. I think that adhesion test should be conducted for different immersion time. For example, Wakai and Harayama clearly showed the adhesion speed of each bacteria in the paper (Materials Technology, 2015, 30.sup5: B38-B43. Doi: /10.1179/1753555714Y.000000220).

Response: The adhesion test was performed for 2 assays after 60 mins and 120 mins immersion and after 2 days (CLSM results). The results after 60 mins were similar to that after 120 mins so this result was not shown. Hence, we did not concentrate on bacterial adhesion rate.

3. Fig. 2: How to determine the top and bottom of biofilm? Authors described about thickness of biofilm without any error values. Are these values from single view? In addition, biofilm in the panel (b) is most ununiformed. If it viewing is correct, adhesion bacterial cells on the surface of SS420 is lowest. Author should show reliable thickness of biofilm with error values based on some observation or viewing.

Response: The CLSM image are in reverse position i.e., the top plane refers to the materials surface. The thicknesses of the biofilms included in the manuscript were the maximum height of the biofilms and 3-4 photos per coupon of the biofilm were taken. The error values of the maximum heights of the biofilm have been added. The main purpose of the CLSM images was to show the distribution of the biofilm. This has been discussed in Result>Adhesion and biofilm formation section.

4. In SEM-EDS analysis, abiotic controls are missing. Especially, observation of SS410 in abiotic condition is significantly important.

Response: The SEM-EDS images have been added to the manuscript. The corrosion rate by weight loss for control condition is updated in Fig. 7. The new added details have been discussed in Results > Corrosion by weight loss and SEM-EDS analysis sections.

5. Fig. 13: I cannot accept this corrosion mechanism. In this paper, authors did not show any evidence supporting this mechanism. Breaking down of passive film is important in the corrosion of stainless steel and this model does not include information of important event. Furthermore, since authors used nutrient rich medium in the corrosion experiment, electron donor for SRB is lactate does not metal. Most recently, Deng et al. reported important of biogenic iron sulfide nanoparticle in MIC by *D. vulgaris*. Please update the proposal mechanism.

Response:

The mechanism presented in Fig.14 in the manuscript was based on previous literature [71, 72]. As per reviewer's recommendation, the breakdown of the passive film steps has been added to Fig. 14. Even though it is nutrient rich medium, electron donor is from both lactate and metal. The reactions can be written as below:

As per data from the paper (<https://doi.org/10.1016/j.ibiod.2014.03.014>), Gibbs free energy of reaction (1) is $\Delta G^\circ = -178$ kJ/mol (at 25°C, pH 7) and of reaction (2) is $\Delta G^\circ = -82.2$ kJ/mol. Thus reaction (1) is thermodynamically more favourable than reaction (2). In other words, iron oxidation is more energetically favourable than lactate oxidation. Therefore, in a solution that contains both lactate and iron, electron donor could come from both lactate and iron. These statements are reflected in the manuscript.

6. SS410 and SS420 are both martensitic stainless steel and chromium content of SS420 is lesser than that of SS410. However, the corrosion of SS410 was sever. Furthermore, the corrosion of austenitic stainless steel SS316 proceeded in the less chromium austenite phase. Could you please discuss it more?

Response:

The reason why SS 410 corroded more than SS 420 is SS 410 had very high bacteria attachment. SS 410 had 2% chromium higher than SS 420. However, the Ni content of SS410 was 0.2% compared to 0.045 for SS420. It appears that Ni seems to have a stronger effect in reducing hydrophobicity value and attracted more bacteria. However, in the case of SS316 although the Ni content was about

10.7% the MIC resistance was greater due to the presence of Mo in stabilising the passive film even though it attracted more bacteria.

For SS 316, the chromium content is considered to be distributed almost uniformly throughout the material (as can be seen from Fig. 10, EDX results). The Cr contents of austenite phases in Fig.10 are not significantly different - 17.9% vs 17.7%. The Cr content of the corroded area was not measured. The points a and b in Fig.10 do not correspond to the corroded areas.

Minor Comments

1. *How to measure hydrophobicity of bacterial cells. Although method of stainless steel was written, those of bacterial cells is missing.*

Response: In the manuscript, we determined the contact angle of the bacteria via bacterial lawn. This method has been indicated in several publications ([0.1016/j.colsurfb.2005.07.020](https://doi.org/10.1016/j.colsurfb.2005.07.020), <https://aem.asm.org/content/48/5/980>) for determining bacteria hydrophobicity behaviour.

2. *Don't use rpm without rotor radius. Author should describe as xg or rpm with rotor radius.*

Response: We have removed the rpm according to your recommendation as we do not know exactly the rotor radius.

3. *Fig. 1: Both panels are "Assay 1". Please change the bottom panel to be "Assay 2".*

Response: We have inserted again the figure for Assay 2.

Reviewer: 2

Comments to the Author(s)

The author in the research paper "A study of bacteria adhesion and microbial corrosion on different stainless steels in environment containing Desulfovibrio vulgaris" they explain to detail the protection of corrosion in different steels using a Gram-negative bacteria.

I enjoyed to read the paper. The results and discussion correspond to the methods and introduction. I have a few comments.

P3L29-32. The author think that the bacterial cell-wall and membrane have an influence in the surface attachment.

Response: Bacterial cell-wall and membrane do have influence in the surface attachment because they determine bacteria hydrophobicity. Several publications indicate this (<https://www.ncbi.nlm.nih.gov/pmc/articles/PMC204020/pdf/aem00125-0181.pdf>).

P3L32-34. D. vulgaris is an aerobic strain... it can growth in low oxygen conditions?

Response: It is obligate anaerobic strain. However, there are some findings show that they can survive under low oxygen conditions as they have function to adapt to it (<https://www.ncbi.nlm.nih.gov/pmc/articles/PMC92413/>, <https://journals.plos.org/plosone/article?id=10.1371/journal.pone.0123455>)

P4L23. Could you add a reference?

Response: The concentration of bacteria was calculated after we added 500 mL of culture medium. So, we cannot provide reference for this.

P4L25-26. Why the authors chose those concentrations?

Response: The intention was to have enough bacteria at the start of the experiment. The bacterial concentration resulted from taking unspecified amount bacteria that were cultured and suspended in nutrient rich sea water. This was to ensure that there were enough bacteria at the start of the experiment. Two dilutions of these were made to get different concentrations.

General

The authors need to homogenise the term hours or hr, using h. Also, N,N-dimethylformamide reagent.

Response: We have corrected this according to your recommendation.

Regarding to adherence. Do the authors know if exist reports about of the gene expression for adhesion to different materials?

Response: We did not study the gene expression; We are aware of some publications about gene expression related to adhesion of bacteria on polymers (<https://doi.org/10.1007/s10856-006-8234-x>) and on titanium alloys (<https://doi.org/10.1016/j.msec.2012.12.063>). This is a very interesting topic beyond the scope of this work.